# GOAL-ORIENTED STATE REDUCTION OF UNKNOWN GAME DYNAMICS TO PRODUCE EFFECTIVE STRATEGIES

## ABSTRACT

Recent reinforcement learning agents perform excellently by generating an internal representation of information crucial for predicting outcomes through huge experience, but do not clarify what essential information (core) is extracted in their representation. A model-based reinforcement learning algorithm, Goal-Oriented Environment Inference (GOEI), has been proposed to solve this issue, and its ability to explicitly learn such core states has been demonstrated in an abstract environment. Here, we validated the ability of GOEI in a more realistic environment, i.e., a competitive card game "Hol's der Geier (The Vulture Gets It)." To our surprise, it achieves a nearly optimal strategy equivalent to the Nash equilibrium by using core states reduced only to 2.9% (452 states) of all possible observations (15,542). These results demonstrate that GOEI effectively excludes information irrelevant to game outcomes, thereby significantly reducing the memory burden.

## 1 INTRODUCTION

Extensive research has been conducted in reinforcement learning(Sutton & Barto, 2018) and game theory (Fudenberg & Tirole, 1991) on algorithms to learn effective strategies to achieve goals in given environments. In particular, decision-making agents using deep neural networks (DNNs) for strategy learning surpass human performance in large-scale tasks such as chess, poker, and video games (Schrittwieser et al., 2020; Brown et al., 2019). Such agents excel at extracting statistical properties from complex observational data through extensive training. However, despite their excellent performance, their decision-making processes rely on intricate internal states and generally lack explainability (Puiutta & Veith, 2020; Robnik-Šikonja & Bohanec, 2018; Verma et al., 2020). Since these agents model all observations that are not necessarily important for reward prediction, their internal models become difficult to interpret. Furthermore, since they are based on offline learning with vast amounts of data, there is much room for improvement in tasks that require online learning to adapt to opponents.

Current approaches to explainable AI (XAI), which aims to analyze information crucial for specific decisions, are typically local and limited in scope, making it hard for humans to comprehend the agent's overall strategy (Lipton, 2018). Simplified surrogate models (Bastani et al., 2017) can approximate specific decision-making scenarios, but they fail to clarify the causal structures underlying tasks. Modeling tasks as Markov decision processes (MDPs) may elucidate such causal structures(Li et al., 2006), but, unlike DNN-based methods, MDPs cannot simplify the state representation efficiently, yielding a surplus number of states to predict irrelevant observations(Takahashi et al., 2024).

Goal-Oriented Environment Inference (GOEI) is proposed to overcome these problems by estimating a minimal state representation (referred to as the "core") necessary for outcome prediction (Takahashi et al., 2024). GOEI exhibits ability of core state extraction to produce an effective strategy in an abstract environment in which cores are explicitly defined but unknown to the agent. This implies that GOEI has the potential to efficiently learn online even in environments with vast observations that are computationally intractable with standard MDP models. However, its validity has yet to be examined in more complex environments.

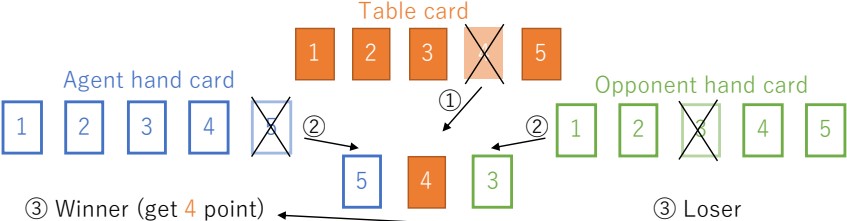

Figure 1: An example of the first round of the game Hol's der Geier (The Vulture Gets It). A card is randomly selected from a set of five table cards and opened on the table (1). The cards selected by individual players from their hands are simultaneously opened (2). The player who selects a card with the highest value among the players earns a score equal to the number on the opened table card (3). The cards used once are discarded and cannot be reused.

Here, we validate GOEI's potential in a realistic, difficult environment for humans to understand what is core, i.e., a competitive card game "Hol's der Geier (The Vulture Gets It)." This game has a complex Nash equilibrium solution, but human players generally employ strategies based on relatively simple state representations. We show that GOEI can reduce state representation to achieve performance equivalent to the Nash equilibrium.

## 2 BACKGROUND

### 2.1 HOL'S DER GEIER (THE VULTURE GETS IT)

Hol's der Geier (The Vulture Gets It) is a multi-player card game in which players compete to win cards opened on the table by playing their set of hand cards. In this study, we consider a two-player version with five cards to win, while a standard set of Hol's der Geier for sale consists of multiple sets of fifteen cards to win for several players.

At the start of each game, each player holds a set of five hand cards numbered 1 through 5. A card is randomly selected from a different set of five cards numbered 1 through 5 and opened on the table. Individual players select respective playing cards from their hands to win the opened table card. The cards selected by the players are simultaneously opened. The player who selects a card with the highest value among the players earns a score equal to the number on the opened table card. If both players play cards with the same number, neither wins a score. This sequence is the first round ($t = 1$, governed by the first table card; Figure 1). The opened table card and the played hand cards are all discarded and cannot be used in subsequent rounds. A similar process is repeated for the remaining table cards in subsequent rounds until all table cards have been opened ($t =$ 2, 3, 4, 5). The player with the highest total score at the end of the game is the winner. Hol's der Geier is a zero-sum game, since one player's gain results in the other player's loss.

To win, players must carefully decide whether to attempt to win the current table card. Higher-value cards are generally advantageous, but each is available only once. Players must predict their opponent's card from observable information that consists of five features: the score difference (SD), the current table card (CT), the agent's remaining hand cards (AH), and the opponent's remaining hand cards (OH), the remaining table cards (RT). The number of possible observations is $28,477$ in total of the rounds. An overwhelming number of observations are given to players. Hence, the reduction of information is required to learn effectively through finite experience.

### 2.2 TYPICAL STRATEGIES AND NASH EQUILIBRIUM

A commonly used simple strategy in human-versus-human games is to play the hand card that matches the current table card ($\pi_0$). A counter strategy often exists against such a deterministic strategy. The counter strategy against $\pi_0$ is to play the current table-card number plus 1, namely, the following card number: $(T \bmod 5)+1$ where $T$ is the current table-card number ($\pi_1$). Similarly, the strategy $\pi_k$: $(T + k - 1 \bmod 5)+1$ is a counter strategy against $\pi_{k-1}$. These strategies only utilize

minimal information from the current table card. Many human players typically play the game by subjectively selecting these simple strategies $\pi_k$.

To overcome a counter strategy, stochastic selection (mixed strategy) is generally effective, because the counter is broken by randomness. The entirely random strategy (Rand) in which selection probabilities are uniform among the available cards can overcome $\pi_1$, $\pi_2$, $\pi_3$, but not $\pi_0$, $\pi_4$. The true optimal strategy is more complicated. Nash equilibrium (NE) among mixed strategies can be calculated, but it is not easy to interpret. There is no strategy to overcome NE against the NE opponent. Therefore, NE is optimal in a certain sense, whereas better strategies exist against a certain fixed strategy other than NE. If an agent based on a reduced state representation could compete equally well against the NE opponent, then the agent is considered to have successfully learned core information to win the game.

## 3   METHODS

### 3.1   STATE REDUCTION PROBLEM IN HOL'S DER GEIER

In this study, we consider a model-based reinforcement learning that explicitly learns state transitions of a given environment by reducing the number of states from redundant observations.

In Hol's der Geier (The Vulture Gets It), the observation $o_t \in O_t$ consists of a combination of five features {SD, CT, AH, OH, RT} (Sec. 2.1). The agent's card selection corresponds to an action $a_t \in A_{o_t}$ at round $t$. The reward $r \in R \equiv \{-1, 0, 1\}$ corresponding to {loss, draw, win} is determined by the score difference at the final round ($t = 5$). The result of the final round is automatically determined by the result of the fourth round ($t = 4$). Therefore, we consider only four rounds ($t = 1, 2, 3, 4$). We supposed the opponent that select a card depending on only $o_t$. Namely, opponent's selection is assumed to be independent of the history in the previous rounds and games. On the assumption, the task for the agent is an episodic reinforcement learning, and the observations satisfy the Markov property

$$P(o_{t+1} \,|\, a_t, o_t, a_{t-1}, o_{t-1}, \cdots) = P(o_{t+1} \,|\, a_t, o_t), \tag{1}$$

where $P(\cdot)$ represents true probability distribution, while the agent model is differently denoted by $p(\cdot)$ in the following.

The optimal strategy can be obtained by selecting the actions that maximize the action-value functions at respective rounds which are obtained as the solution to the optimal Bellman equation (Sutton & Barto, 2018),

$$Q(a_t, o_t) \;=\; \sum_{o_{t+1}} \left\{ \max_{a_{t+1}} Q(a_{t+1}, o_{t+1}) \right\} P(o_{t+1} \,|\, a_t, o_t),$$

$$Q(a_4, o_4) \;=\; \sum_r r P(r \,|\, a_4, o_4). \tag{2}$$

In this game, however, too many obsevations exist (see Sec. 2.1), and it is practically difficult to solve (2) with inferring $P(o_{t+1} \,|\, a_t, o_t)$. One practical solution is Q-learning in which the action values are directly estimated online,

$$q(a_t, o_t) \;\leftarrow\; (1 - \eta) \, q(a_t, o_t) + \eta \max_{a_{t+1}} q(a_{t+1}, o_{t+1}),$$

$$q(a_4, o_4) \;\leftarrow\; (1 - \eta) \, q(a_4, o_4) + \eta \, r, \tag{3}$$

where $\eta$ is the learning rate. Q-learning works within linear memory capacity for the number of observations. However, it extracts no transition rule of environment. In order to extract the transition rule to optimize the strategy, an effective method for state reduction from redundant observations is required.

To select the optimal action at round $t$, it is necessary to predict the future reward for any following action sequence $\mathcal{A}_{t:4} \equiv [a_t, a_{t+1}, \cdots, a_4]$. The reduced state representation should be sufficient for the reward prediction. The state reduction model $p(s_t \,|\, o_t)$, the state transition models $p(s_{t+1} \,|\, a_t, s_t)$, and $p(r \,|\, a_4, s_4)$ ($s_t \in S$) should satisfy

$$\forall t < 4, \; \forall \mathcal{A}_{t:4}, \;\; P(r \,|\, \mathcal{A}_{t:4}, o_t) = \sum_{s_t, \cdots, s_4} p(r \,|\, a_4, s_4) \prod_{\tau=t}^{3} p(s_{\tau+1} \,|\, a_\tau, s_\tau) \, p(s_t \,|\, o_t). \tag{4}$$

The state reduction problem is to infer the models $p(s_t \mid o_t)$, $p(s_{t+1} \mid a_t, s_t)$ and $p(r \mid a_4, s_4)$ for the state set with the minimal size $|S|$ satisfying (4) (Takahashi et al., 2024). The minimal state set is referred to as "core."

## 3.2 GOAL-ORIENTED ENVIRONMENT INFERENCE

"Goal-oriented environment inference (GOEI)," is a concrete algorithm proposed to solve the state reduction problem (Sec. 3.1) (Takahashi et al., 2024).

The condition (4) should hold at all rounds $t = 1, \cdots, 4$. However, the generative graphs at different rounds $t$ conflict: the state $s_t$ is generated from the observation $o_t$ for the condition at $t$, while $s_t$ is generated through the state transition $p(s_t \mid s_{t-1}, a_{t-1})$. To resolve this conflict, GOEI once infers the different model $p(a_t \mid s_{t+1}, s_t)$ keeping the correlation structure of the original $p(s_{t+1} \mid a_t, s_t)$, and later infers the original model $p(s_{t+1} \mid a_t, s_t)$ for each transition by using the inferred states $s_t$, $s_{t+1}$ and the actual action $a_t$. This approach is ensured by Bayes theory in which Bayesian inference is independent of the true causal direction.

The inference model of GOEI incorporating the episodic structure of the game is described as

$$P(r, \mathcal{A}, \mathcal{S}, \Theta \mid \mathcal{O}) = p(r \mid s_4, a_4, \Theta^R) \prod_{t=2}^{3} p(a_{t-1} \mid s_t, s_{t-1}, \Theta_t^A) \, p(s_t \mid o_t, \Theta_t^S) p(\Theta), \quad (5)$$

where, $\Theta \equiv [\Theta_2^S, \ldots, \Theta_4^S, \Theta_1^A, \ldots, \Theta_3^A, \Theta^R]$ represent the model parameters of respective probability distributions. Additionally, since the first observation $o_1$ only contains the pattern of the current table card (5 patterns), no state reduction is applied at this round, and $s_1 = o_1$.

In Hol's der Geier (The Vulture Gets It), observations, states, rewards, and actions are all discrete variables. Therefore, all the probability distributions are modeled using categorical distributions. The model parameters are assigned Dirichlet distributions, which are the conjugate priors for categorical distributions.

To facilitate state reduction, the model parameters for the clustering rule are assumed to follow a Dirichlet process, which can handle an indefinite number of states. The posterior distribution of the model parameters is approximated with approximated probability distribution $q(\cdot)$ using variational Bayesian inference:

$$q(\Theta) \, q(S) \approx P(\Theta, \mathcal{S} \mid r, \mathcal{A}, \mathcal{O}). \quad (6)$$

Variational Bayesian inference approximates the posterior distribution by maximizing the evidence lower bound (ELBO). When the Dirichlet process is applied to the clustering rule, models with fewer states achieve higher ELBO values. As a result, variational Bayesian inference prefers models with fewer states when their performance is equivalent.

We can express the approximated probability distribution $q$ as a combination of three components: the probability of being in state $s$ at round $t$ given observation $o$; the probability of taking action $a$ at round $t$ given that the agent was in state $s$ at round $t-1$ and transitioned to state $s'$; and the probability of receiving a reward $r$ at round 4 given that the agent was in state $s$ and took action $a$. This can be represented by the following equation:

$$q(\Theta) = \prod q_{s,a}^R(\Theta_{s,a}^R) \, q_{t,o}(\Theta_{t,o}^S) \, q_{t,s,s'}(\Theta_{t,s,s'}^A). \quad (7)$$

The probability of state s is modeled using a Dirichlet process, while the probabilities for actions and rewards are modeled using a Dirichlet distribution:

$$q_{t,o}^{(i)}(\Theta_{t,o}^S) = \text{DP}\left(\Theta_{t,o}^S \,\middle|\, \alpha, a_{t,o}^{(i)}\right),$$

$$q_{t,s,s'}^{(i)}(\Theta_{t,s,s'}^A) = \text{DD}\left(\Theta_{t,s,s'}^A \,\middle|\, b_{t,s,s'}^{(i)}\right), \qquad q_{s,a}^{(i)}(\Theta_{s,a}^R) = \text{DD}\left(\Theta_{s,a}^R \,\middle|\, c_{s,a}^{(i)}\right), \quad (8)$$

where $a$, $b$, and $c$ are the respective hyperparameters. We define the initial values using $\alpha$, $\beta$, $\gamma$ and a small amount of noise, $\varepsilon$, as follows:

$$a_{t,o}^{(0)} = \varepsilon, \qquad b_{t,s,s'}^{(0)} = \beta + \varepsilon, \qquad c_{s,a}^{(0)} = \gamma + \varepsilon. \quad (9)$$

Here, $\mathrm{DP}(x \mid \alpha, a)$ indicates that a new state will appear with a probability of $\alpha/(\sum a_j + \alpha)$, while an existing state $j$ will appear with a probability of a $a_j/(\sum a_j + \alpha)$, where a is a vector of observed counts $a_j$.

In the learning process for each round, the procedure of first calculating $q(s)$ based on the approximated parameter distribution $q(\theta)$ and then, conversely, updating $q(\theta)$ from $q(s)$is repeated until the model evidence divided by the number of data points ceases to improve for a certain number of times, known as patience. In this paper, we set the tolerance (tol) to $10^{-5}$ and patience to 10. Across all experiments, the actual number of iterations ranged from approximately 20 to 50.

For the overall process, we first calculate the approximated distributions for $\Theta_4^S$, $\Theta^R$, and $s_4$. Next, we use the calculated probabilities of $s_4$ to find $\Theta_3^S$, $\Theta_4^A$, and $s_3$. We then similarly determine $\Theta_2^S$, $\Theta_3^A$, and $s_2$, and finally find $\Theta_2^A$, and $s_1$.

## 3.3 EVALUATION OF ENVIRONMENT INFERENCE

The model-based reinforcement learning is separated into two tasks: environment inference and strategy optimization. Both generally interact in online learning because strategy changes affect experiences obtainable for environment inference. To evaluate the performance of GOEI purely in environment inference, we separated the inference learning from the performance test. We trained GOEI and Q-learning to learn through a set of games between two fixed strategies, the random strategy (Rand) versus the Nash equilibrium (NE) strategy. Separately, we tested how much reward would be obtained against the NE opponent, if the inferred model of GOEI at every epoch was used to select optimal actions through the optimal Bellman equation. In the same way, we evaluated the performance of Q-learning against the NE opponent in the separate test of the greedy selection based on the inferred action values at every epoch.

We prepared a training set of 300,000 games divided into 3,000 epochs (100 games per epoch). We allowed the inference agents (GOEI and Q-learning) to learn games both from the Rand player and from the NE player. Namely, 200 games were used for training per epoch. At each epoch, GOEI updated the models through the variational Bayes procedure for the pooled 200 games by using the result of the previous epoch as the priors.

We iterated the games 10,000 times per epoch and calculated the reward rate for the performance test. During the performance test, the inference learning of GOEI and Q-learning was stopped. We averaged the reward rates across epochs 1 to 3,000, and the average reward rate across 3,000 epochs was used to evaluate the performance on each training set. We iterated training of environment inference for each training set 21 times with different seeds, and calculated the median and quartiles of the average reward rates.

The degree of state reduction is also important for the evaluation of GOEI performance. We define the representative state $s_t^*$ per observation $o_t \in O_t$ to maximize the state probability $p(s_t|o_t)$ in the state reduction model. Then, the representative for either observation constitutes a set of representative states, $S_t^* \subset S_t$. We evaluated the performance of state reduction by comparing the number of representative states, $|S_t^*|$, with that of observations, $|O_t|$. In games of Rand vs. NE, the number of possible observations is restricted to $15,542$ ($|O_1| = 5, |O_2| = 300, |O_3| = 4,209, |O_4| = 11,028$) because of action sequences never caused by the NE strategy.

We also define another evaluation method with entropy $H(S_t)$ to incorporate the whole probability distribution,

$$H(S_t) = -\sum_{s_t \in S_t} P(s_t) \log P(s_t), \ \ P(s_t) = \sum_{o_t \in O_t} p(s_t \mid o_t) P(o_t), \tag{10}$$

where $P(o_t)$ is estimated from training sets of games by Rand vs. NE. We note that for a uniform distribution, $e^{H(S)}$ gives the number of states. We also calculated the median and quartiles of these indices $|S_t^*|$ and $e^{H(S)}$ for 21 times trainings. We described the only median at 3,000 epoch on Table 1.

For comparison, we also calculated the effective state number of NE. The NE strategy is described as the action probability distributions conditioned on respective observations. Each of them can be calculated by defining a state representing a set of equal expected rewards earned with players' actions. At round $t = 4$, rewards for actions are self-evident, as the game's outcome is automatically

Table 1: Performances against the NE strategy and degrees of state reductions.

| | | | | $|O_2|$ | $e^{H(O_2)}$ | $|O_3|$ | $e^{H(O_3)}$ | $|O_4|$ | $e^{H(O_4)}$ |
|---|---|---|---|---|---|---|---|---|---|
| | | | Observation | 300 | 193.5 | 4209 | 2417.6 | 11028 | 6043.6 |
| | Reward rate (epoch 1~3,000) | | | State representation size (at epoch 3,000) | | | | | |
| | 25% | 50% | 75% | $|S_2^*|$ | $e^{H(S_2)}$ | $|S_3^*|$ | $e^{H(S_3)}$ | $|S_4^*|$ | $e^{H(S_4)}$ |
| NE | 0.000 | 0.000 | 0.000 | 247 | 164.6 | 945 | 303.6 | 69 | 16.8 |
| Rand | −0.527 | −0.527 | −0.527 | 5 | 3.8 | 10 | 8.1 | 10 | 8.4 |
| $\pi_0$ | −0.125 | −0.125 | −0.125 | 5 | 5.0 | 5 | 5.0 | 5 | 5.0 |
| QL ($\eta$) | | | | | | | | | |
| 0.05 | −0.086 | −0.083 | −0.080 | - | - | - | - | - | - |
| 0.10 | −0.081 | −0.080 | −0.078 | - | - | - | - | - | - |
| *0.20 | −0.081 | −0.079 | −0.078 | - | - | - | - | - | - |
| 0.50 | −0.104 | −0.103 | −0.102 | - | - | - | - | - | - |
| GOEI $(\beta, \alpha)$ | | | | | | | | | |
| 0.1, 11 | −0.084 | −0.073 | −0.053 | 32 | 34.2 | 56 | 40.0 | 263 | 131.3 |
| 0.1, 25 | −0.042 | −0.028 | −0.019 | 12 | 21.1 | 16 | 14.8 | 304 | 148.1 |
| 0.1, 50 | −0.026 | −0.016 | −0.014 | 15 | 32.5 | 12 | 11.5 | 363 | 160.0 |
| 0.2, 11 | −0.062 | −0.044 | −0.035 | 26 | 23.4 | 28 | 22.3 | 423 | 233.7 |
| *0.2, 25 | −0.012 | **−0.010** | −0.009 | 8 | 15.0 | 31 | 27.0 | 408 | 198.8 |
| 0.2, 50 | −0.017 | −0.015 | −0.013 | 14 | 40.9 | 30 | 52.9 | 431 | 204.6 |
| 0.3, 11 | −0.071 | −0.059 | −0.048 | 68 | 63.2 | 62 | 39.5 | 426 | 225.9 |
| 0.3, 25 | −0.038 | −0.028 | −0.018 | 111 | 249.7 | 23 | 35.8 | 398 | 197.1 |
| 0.3, 50 | −0.081 | −0.071 | −0.061 | 92 | 302.1 | 142 | 331.8 | 413 | 196.9 |

determined by the cards played. At $t = 1, 2$, and 3, rewards for actions are defined by the rewards obtained from the subsequent actions following the NE in round $t + 1$. The number of the states for the calculation of NE is used for the state representation size, $|S^*|$, of NE. For the states, the entropy $H(S_t)$ is defined as in eq. (10).

## 4 RESULTS

### 4.1 PERFORMANCE OF GOEI COMPARED WITH Q-LEARNING

We trained GOEI at different parameters $\alpha$ for the Dirichlet process (DP) and $\beta$ for the Dirichlet distribution (DD). We found that the best performance was obtained at $\beta = 0.2$ and $\alpha = 25$ (Table 1). For comparison, we trained Q-learning at different learning rates $\eta = 0.05, 0.1, 0.2, 0.5$ and found the best performance at $\eta = 0.2$. Figure 2A shows the learning curves of GOEI and Q-learning at their best parameters. GOEI performed surprisingly well compared to the simple strategy $\pi_0$ and Q-learning. Moreover, GOEI could rapidly learn a good strategy that is almost comparable with the NE opponent, with the reward rate $\simeq 0$. In contrast, Q-learning required significantly more epochs than GOEI to increase the reward rate.

The nearly optimal strategy of GOEI could arise owing to its extremely compact state representations. The number of states was much reduced compared to that of observations, $|O_t|$, at each round $t = 2, 3, 4$ (Fig.2B). At rounds $t = 2$ and 3, the number of states was even smaller than that of the NE strategy (dashed lines in Fig. 2B). The poor performance of Q-learning indicates that the number of observations is too large even for the simple Q-learning algorithm. Altogether, our results demonstrated the importance of state reduction in a competition game accompanied by huge observations and against strong opponents.

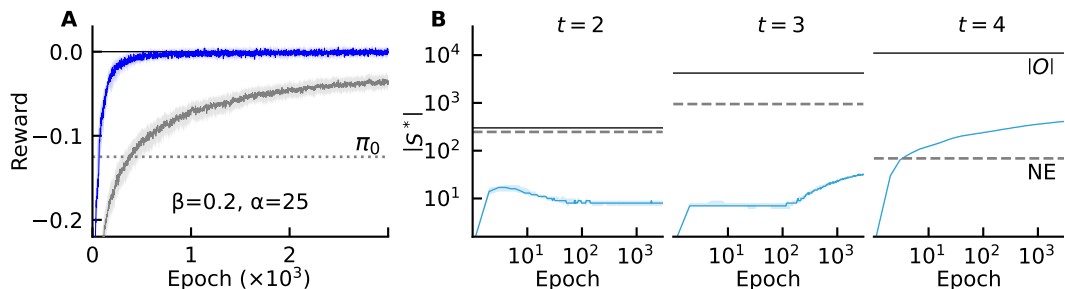

Figure 2: Performance of GOEI. A, The learning curves of GOEI (blue) and Q-learning (grey) at the best parameters. The solid lines and transparent regions respectively show the medians and quartile ranges of reward rates in the separate performance tests against the NE opponent per epoch. The dashed line indicates the reward rate of the simple strategy $\pi_0$. B, The time course of the number of states in GOEI at each round, $|S_2^*|$, $|S_3^*|$, and $|S_4^*|$, is shown by the median and quartile range, as in A. Solid and dashed grey lines indicate the number of observations and that of states in NE, respectively. To expand early phase of learning, the horizontal axes are shown on log scales.

## 4.2 CONTENT OF REMAINING INFORMATION

GOEI could reduce the state representation, while preserving the information necessary to produce a near-optimal strategy. To clarify what information content remained in the reduced state representation, we examined the degree of state reduction for the following features of observations: score difference (SD), current table card (CT), agent's remaining hand (AH), opponent's remaining hand (OH), and remaining table cards (RT). The value of each feature, $o_t^F \in O_t^F$, $F \in \{\text{SD, CT, AH, OH, RT}\}$, is determined by the observation $o_t \in O_t$ through each map $O_t \mapsto O_t^F$. We evaluated the mutual information between the state and each feature, $I(O_t^F; S_t) = H(O_t^F) - H(O_t^F|S_t)$, where the entropy of the feature $H(O_t^F)$ gives the upper bound of the mutual information, $I(O_t^F; S_t) \leq H(O_t^F)$. Therefore, the conditional entropy $H(O_t^F | S_t)$ is interpreted as information loss through the state reduction. We estimated the mutual information by using the joint probability distribution $P(o_t^F, s_t)$ calculated with the state reduction model $p(s_t|o_t)$ of GOEI and the observation probability distribution $P(o_t)$ estimated from the training sets of games by Rand vs. NE,

$$P(o_t^F, s_t) = \sum_{o_t \in O_t} P(o_t^F \,|\, o_t) p(s_t \,|\, o_t) P(o_t), \tag{11}$$

where $P(o_t^F \,|\, o_t)$ is a deterministic (one-hot) probability distribution corresponding to the map $O_t \mapsto O_t^F$.

After the state reduction, the information loss (red bars in Figure 3) occupies a large portion of the entropy, and the mutual information (blue bars) is very low for each feature, implying that overall GOEI reduced significant portions of information about the individual features. However, we may say that, compared to the other features, information about the current table card CT and the remaining table cards RT was relatively preserved at rounds $t = 2$ and $t = 3$. In contrast, information on the score difference SD was preserved at round $t = 4$, as indicated by non-negligible mutual information. This seems reasonable because SD is unlikely to be important in early rounds but becomes crucial as the game approaches the final round. Information about the agent's hand AH and the opponent's hand OH was almost completely reduced throughout the game. However, these pieces of information are likely to be crucial for learning a near-optimal strategy. Altogether, these results suggest that the required information is maintained in complex combinations of all the features.

## 4.3 INFLUENTIAL PARAMETERS ON LEARNING

We examined how the parameter values influences the performance of GOEI. The parameter $\beta$ of the Dirichlet distribution sets the prior distribution for the transition rules. The Dirichlet distribution coincides with a uniform distribution for $\beta = 1$, while it tends to generate sparse, one-hot-like states as $\beta \to 0$. These properties lead us to the following speculation: at small values of $\beta$, GOEI is

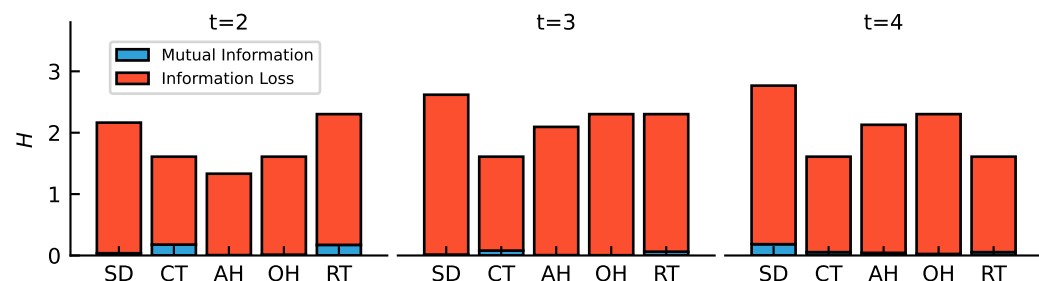

Figure 3: Mutual information $I(O_t^F; S_t) = H(O_t^F) - H(O_t^F|S_t)$ (blue) and information loss $H(O_t^F|S_t)$ (red) were shown as stacked bar charts for each observable feature: $F \in \{$score difference (SD), current table card (CT), agent's remaining hand (AH), opponent's remaining hand (OH), remaining table cards (RT)$\}$ at rounds $t =$ 2, 3, 4 after 3,000 epoch training. Note that each total bar length (blue + red) indicates $H(O_t^F) = I(O_t^F; S_t) + H(O_t^F|S_t)$, which gives the upper bound of $I(O_t^F; S_t)$.

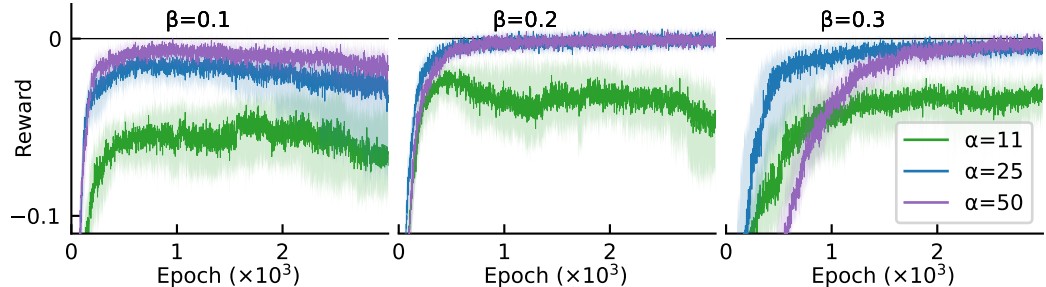

Figure 4: Comparison of GOEI's performance at several parameter values ($\alpha =$ 11, 25, 50; $\beta =$ 0.1, 0.2, 0.3) shown as in Figure 2A.

sensitive to the initial observational data, which may accelerate learning but may generate instability and convergence to local minima.

The parameter $\alpha$ sets a clustering rule for of the Dirichlet process. The larger the value of $\alpha$, the higher the probability of generating a novel state different from the existing ones. Therefore, we speculate that a larger $\alpha$ enhances exploration and helps avoid local minima, but it may increase the number of states and slow down learning.

We confirmed that our speculations were indeed the cases (Figure 4). GOEI could earned rewards more rapidly at $\beta = 0.2$ than at $\beta = 0.3$. In contrast, at $\beta = 0.1$, GOEI became unstable at later epochs, although its behavior was initially similar to that for $\beta = 0.2$. Such a trend holds for the parameter $\alpha$. GOEI could earned rewards more rapidly at $\alpha = 25$ than at $\alpha = 50$, specially at $\beta = 0.3$. In contrast, learning at $\alpha = 11$ reached only poor performance.

## 5 SUMMARY AND DISCUSSION

In this study, we explored a method to obtain concise representations of game strategies using GOEI. Our results demonstrated that GOEI acquires a near-optimal strategy over an extremely concise state representation deduced from a large number of observations. The number of states reduced by GOEI at the best parameters was $452 = 5 + 8 + 31 + 408$ (median) in total through rounds $t = 1, 2, 3, 4$ at epoch 3,000 (Table 1). This is only 2.9% of the number of observations ($15,542 = 5 + 300 + 4,209 + 11,028$), while the median of performance at 3,000 epochs was indistinguishable from the optimal one ($\simeq 0$; Figure 2A). Thus, GOEI provides a dynamic information reduction method that is not only effective for abstract environments but also for complex competitive games.

Here, we adopted the five-card version of Hol's der Geier due to the limitation of the memory capacity of our computing resources (NVIDIA RTX4080 SUPER with 12GB memory). GOEI could apply

to the five-card version due to its excellent state number control provided by the Dirichlet process. After learning, GOEI successfully reduced the number of states without sacrificing its performance. The Dirichlet process can handle an explicit upper bound for variable size, as in the present case. Therefore, GOEI may apply to versions with cards more than five by introducing an appropriate upper bound. This property of GOEI may enable it to solve various problems more complex than Hol's der Geier. The evaluation of GOEI's performance under memory capacity constraints is open to future studies.

There are, however, some limitations in the current method. To evaluate the performance of solely environment inference, we separated environment inference and strategy optimization by training GOEI through games between fixed strategies: Rand vs. NE. In normal human-playing situations, however, environment inference and strategy update are simultaneously executed. Learning improves inferences and updates the strategy, but the strategy change affects future experiences used in inferences. In such interactive learning, environment inference and strategy optimization may interfere with each other. In particular, available experience may be restricted in games with a large number of possible observations, such as Hol's der Geier, which can cause a critical problem for inference. The effectiveness of the GOEI function in interactive learning should be further confirmed.

Our original motivation for developing GOEI was to improve the explainability by reducing state representations. GOEI succeeded in greatly reducing state representations and extracting transition rules on the reduced states. However, we could not give a verbal explanation of the reduced state representation more concretely than Figure 3. State reduction may be necessary for explainability, but it does not always lead to a concrete explanation. The extent to which reduced representations contribute to explainability is yet to be clarified.

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
