# OpenReview forum: "Goal-oriented state reduction of unknown game dynamics to produce effective strategies"
_ICLR.cc/2026/Conference — Submitted to ICLR 2026_

### Official Review · Reviewer_PD4Z · 2025-10-25

**Soundness:** 1
**Presentation:** 2
**Contribution:** 1
**Rating:** 0
**Confidence:** 4

**Summary:**

This paper explores application of Goal-Oriented Environment Inference to state space reduction problem in card boardgames, especially in two-player game called "Hol’s der Geier (The Vulture Gets It)”. Authors test how well will GOEI perform against Nash equilibrium random opponents under different set of hyperparameters.

**Strengths:**

Experiment that were conducted on "Hol’s der Geier (The Vulture Gets It)” are solid and shows results for different sets of hyperparameters as well as performance trends for these sets.

**Weaknesses:**

- Work uses only single environment to conduct its experiments, which do not provide any evidence on how GOEI will perform on other environments of the same type or other types of environments.
- Conducted experiments compare GOEI only with DQN out of all RL algorithms. While GOEI is model-based RL algorithm and especially focused on state representation learning, it's crucial to compare it with other MBRL algorithms (i.e. [Dreamer v3](https://arxiv.org/abs/2301.04104), [DreamerPRO](https://proceedings.mlr.press/v162/deng22a.html) and others)
- Work do not propose either novel algorithms nor experimental benchmarks. The algorithm (GOEI) were already [published](https://doi.org/10.1016/j.neunet.2024.106246) and used as-is with citation.

**Questions:**

- What is the contribution of the paper aside from applying GOEI to "The Vulture Gets It"?
- Is there any experiments conducted on other environments or comparison with other model-based algorithims?

---

### Official Review · Reviewer_sNkK · 2025-10-31

**Soundness:** 3
**Presentation:** 3
**Contribution:** 1
**Rating:** 2
**Confidence:** 3

**Summary:**

The paper investigates Goal-Oriented Environment Inference (GOEI), a model-based reinforcement learning algorithm designed to extract minimal, goal-relevant state representations (“core states”) from redundant observations while preserving the agent’s ability to predict outcomes. Previous work demonstrated GOEI’s capacity for abstraction in simplified environments.
This paper extends GOEI to a realistic competitive game setting, specifically the two-player card game Hol’s der Geier (The Vulture Gets It). The authors show that GOEI can reduce the number of state representations to only 2.9% of all possible observations while achieving a near-Nash-equilibrium performance, outperforming Q-learning in both efficiency and interpretability. The results suggest that GOEI can identify causally important features and drastically reduce representational complexity in competitive settings.

**Strengths:**

1. Demonstrates GOEI’s effectiveness on a non-trivial, stochastic, competitive game
2. Presents clear comparative experiments with baselines such as Q-learning and Nash equilibrium strategies, along with ablations over hyperparameters (α, β).
3. The evaluation includes entropy-based measures and mutual information analyses, providing a multi-faceted view of representational compactness.

**Weaknesses:**

1. The evperiment is confined to a simplified (five-card) version of Hol’s der Geier, which restricts claims about scalability and general applicability to larger, more complex environments.
2. While GOEI achieves state compression, the interpretability of the resulting “core states” remains opaque. The paper acknowledges this but does not propose methods to make reduced representations human-comprehensible.
3. While the experimental validation is thorough and provides useful empirical insights, this work is primarily an application and validation of the existing GOEI method in a new, more realistic environment (the competitive card game Hol’s der Geier). So it is lack of methodological novelty.

**Questions:**

No question for the work done in this paper.

---

### Official Review · Reviewer_dA92 · 2025-11-01

**Soundness:** 2
**Presentation:** 1
**Contribution:** 2
**Rating:** 2
**Confidence:** 3

**Summary:**

The reviewed work evaluates a model-based reinforcement learning algorithm, "Goal-Oriented Environment Inference" on a two player zero sum game called "Hol's der Geier" (The Vulture Gets It).

Much of the paper is concerned with explaining the game and common strategies for it. Then, it quickly introduces the problem of state reduction and describes the previously introduced Goal-oriented environment inference (GOEI) methodology.

**Strengths:**

- The authors seem to do a thorough job in investigating the dependence of the proposed methodology on the different parameters.

**Weaknesses:**

- The writing is overall not sufficient to convey the technical ideas. While a lot of real estate is spent on introductoy material, the core of the evaluated methodology is introduced hastily and with excessive amounts of vague language ("this approach is ensurred by Bayes theory in which Bayesian
inference is independent of the true causal direction." or "where P (·) represents true probability distribution, while the agent model is differently denoted by p(·) in the following."

- The evaluation is limited, partly by being restricted to just a single game and partly by claiming success by virtue of achieving nominal performance only when playing against the Nash Equilibrium

- Even if it were executed perfectly, the contribution of investigating the performance of a known method on a single new game seems insufficient for an ICLR paper.

**Questions:**

1. What do you mean by "this approach is ensurred by Bayes theory in which Bayesian
inference is independent of the true causal direction." ?

2. What exactly is $p(s_t, o_t)$? What is "The state reduction model'?

3. You write
"If an agent based on a reduced state representation could compete equally
well against the NE opponent, then the agent is considered to have successfully learned core information to win the game."

Many Nash Equilibrium strategies amount to neutralizing the opponent's agency over the average outcome of the game (think, the 50-50 strategy in matching pennies). Thus, this claim seems fundamentally unjustified.

---

### Meta-Review · Area_Chair_Ci67 · 2026-01-05

**Summary:**

The reviewers raised consistent and significant concerns, reaching a clear consensus toward rejection. The primary issue is the lack of contribution and novelty. The experimental evaluation is narrow, limited to a simplified version of a single game, and the work lacks sufficient new algorithms, theoretical insights, or benchmarks.

No rebuttal was provided to address these concerns. Based on the reviewers’ assessments, the AC recommends rejection.

**Reviewer Concerns:**

No rebuttal was provided.

**Reviewer Scores:**

No rebuttal was provided. The reviewer may not change their scores.

---

### Decision · Program_Chairs · 2026-01-26

Reject